# Exploring the Relationship between Medical Research Literacy and Respondents’ Expressed Likelihood to Participate in a Clinical Trial

**DOI:** 10.3390/ijerph192215168

**Published:** 2022-11-17

**Authors:** Jennifer Dykema, Cameron P. Jones, Dana Garbarski, Mia Farias, Dorothy Farrar Edwards

**Affiliations:** 1Department of Sociology, University of Wisconsin-Madison, Madison, WI 53706, USA; 2University of Wisconsin Survey Center, University of Wisconsin-Madison, Madison, WI 53706, USA; 3Department of Statistics, University of Wisconsin-Madison, Madison, WI 53706, USA; 4Department of Sociology, Loyola University Chicago, Chicago, IL 60660, USA; 5Departments of Kinesiology and Medicine, University of Wisconsin-Madison, Madison, WI 53706, USA

**Keywords:** medical research literacy, health literacy, medical research participation, clinical trials, survey measurement

## Abstract

Medical research literacy (MRL) is a facet of health literacy that measures a person’s understanding of informed consent and other aspects of participation in medical research. While existing research on MRL is limited, there are reasons to believe MRL may be associated with a willingness to participate in medical research. We use data from a racially balanced sample of survey respondents (n = 410): (1) to analyze how MRL scores vary by respondents’ socio-demographic characteristics; (2) to examine how MRL relates to respondents’ expressed likelihood to participate in a clinical trial; and (3) to provide considerations on the measurement of MRL. The results indicate no differences in MRL scores by race or gender; younger (*p* < 0.05) and more educated (*p* < 0.001) individuals have significantly higher MRL scores. Further, higher MRL scores are associated with significantly lower levels of expressed likelihood to participate in a clinical trial. Additionally, the MRL scale included both true and false statements, and analyses demonstrate significant differences in how these relate to outcomes. Altogether, the results signal that further research is needed to understand MRL and how it relates to socio-demographic characteristics associated with research participation and can be measured effectively.

## 1. Introduction

### 1.1. Background and Past Research

Medical research literacy (MRL) refers to an individual’s understanding of research principles and procedures as well as their ability to interpret and understand the essential elements of informed consent related to participation in medical research studies; for example, that participation should be voluntary and that the goals of medical research are not the same as medical care [1,2]. Medical research literacy is closely related to the concept of health literacy, defined by the US Centers for Disease Control and Prevention as the degree to which individuals have the ability to find, understand, and use information to inform health-related decisions for themselves and others [3]. The concept of MRL affords an opportunity to contribute to the larger discussion about the public’s perceptions and understanding of the research process more generally [4]. For example, are people with higher MRL more, less, or equally likely to participate in medical research studies?

MRL is also situated at the intersection of health literacy and science literacy, and research exists stressing societal needs to understand and reliably measure both concepts. Science literacy refers to an individual’s understanding of scientific concepts and processes. Ploomipuu and colleagues [5] wrote about “the need to promote health literacy as the goal of health education for all, at all educational levels,” while Plohl and Musil [6] and Paasche-Orlow et al. [7] found a positive relationship between health literacy and better health outcomes. Serpa et al. [8] noted the “need to promote scientific literacy in the general population” during the midst of misinformation during the COVID-19 pandemic. Beyond this baseline need, Allum et al. [9] found significant racial and ethnic disparities in science literacy. 

Past research also exists on the relationship between health literacy and an individual’s likelihood to participate in medical research. Kripalani et al. [10] reported a positive relationship between health literacy and both interest in and actual participation in medical research [11]. Ousseine et al. [12] isolated health literacy as a predictor of being invited to and enrolling in a clinical trial, finding subjects with lower literacy levels were less likely to report having been invited to participate in a clinical trial in the past, but health literacy did not impact enrollment rates once invited. The relationship between health and science literacy and participation in medical research suggests that a similar relationship may exist with medical research literacy. MRL likely affects the decision-making process individuals undertake when considering participating in medical research, as discussed later in this introduction. Furthermore, from the established evidence, MRL levels may be problematically low in the general population and additionally may experience significant racial and ethnic disparities.

A substantial body of research shows that underrepresented groups are less likely to participate in clinical trials [13]. The National Institutes of Health (NIH) has mandated full inclusion of historically underrepresented groups in federally funded research [14]. The NIH considers the following ethnoracial groups as underrepresented in biomedical research: Individuals from racial and ethnic groups such as Blacks or African Americans, Hispanics or Latinos, American Indians or Alaska Natives, Native Hawaiians, and other Pacific Islanders. Underrepresentation prevents advancements in the prevention, diagnosis, and treatment of diseases and disabling conditions from being generalized to underrepresented groups, which sustains significant health disparities. Despite efforts to increase the inclusion of minoritized populations, non-White populations bear the burden of disease but usually do not reap the benefits of research advancements [15]. Given the likely effect of MRL on willingness to participate in medical research, it is important to understand how MRL varies within a sample of individuals based on their ethnoracial identification, gender, education, and age. A better understanding of these relationships may elucidate one of the causes of the underrepresentation addressed by the NIH.

At the center of the informed consent process is the agreement between the researcher and the individual being studied. More research is needed to understand the process an individual undergoes when making the decision to participate or not and how to best inform potential participants [16]. It is likely that some combination of MRL and trust in, past experiences with, and perceptions of medical research moderates the decision-making process. While these relationships are worthy of research, literature on the effect of MRL is perhaps the most limited. For example, while Brody et al. [1] demonstrated the positive effect of research literacy among parents enrolling their children in research studies, the relationship between MRL and the intent to participate among a general population of adults is yet to be examined.

Finally, standardized measurement is a key component of replicating and verifying scientific findings [17]. While there are no standardized instruments to measure MRL, previous work assessing this concept has made use of true and false statements to capture how knowledgeably individuals understand some of the basic elements of informed consent [1]. A large body of research indicates true-false statements can be used to effectively measure an individual’s knowledge about a subject, but we are unaware of research that probes this issue in depth for the measurement of MRL. 

### 1.2. Current Study and Research Questions

This exploratory study examines the relationships among respondents’ MRL, their socio-demographic characteristics, and their expressed likelihood to participate in a clinical trial. Further, due to systematic differences in how respondents answered MRL questions that were presented as true statements instead of false statements, we also examine how the true versus false presentation affects the measurement properties of MRL.

Our research questions are:Does MRL vary by respondents’ socio-demographic characteristics, including their ethnoracial identification, gender, education, and age?Is MRL related to respondents’ expressed likelihood to participate in a clinical trial?Are these relationships dependent on whether a statement is presented as true or false?

## 2. Materials and Methods

### 2.1. Study Design and Sample

Data for this study are from the Voices Heard computer-assisted telephone interview survey [18], which was designed to measure perceptions of barriers and facilitators to participating in medical research studies that collect biomarkers (e.g., saliva and blood) among respondents from four groups defined by their ethnoracial identification (i.e., Black, Latino, American Indian, and White) [19]. We employed a quota sampling strategy because screening to identify members in non-White groups would have been prohibitively expensive. The quota sample consisted primarily of volunteers but also used a targeted list of names provided by a commercial vendor. Interviewers conducted 410 usable interviews (in English only) with an average length of 25.21 min between October 2013 and March 2014. The respondents received a USD 20 cash incentive.

The questions in the survey were informed by cognitive interviews conducted with Black, Latino, American Indian, and White participants [20]. In the cognitive interviews, the participants were asked questions being tested for use in the telephone interview [21]. After the participants provided responses to the closed-ended survey questions, the interviewers administered a series of structured, open-ended probes and follow-up questions designed to uncover how the participants formulated their answers, to reveal any problems they had with comprehension of specific terms or retrieval of information from memory, and to the document issues that the participants demonstrated mapping their responses onto the response categories. An analysis of the data from the cognitive interviews did not reveal any evidence of systematic differences across ethnoracial groups in their comprehension of the questions.

The final set of 96 questions included in the survey asked about: the likelihood of participating in medical research based on the type of study (e.g., to collect tissue for a clinical trial) and the characteristics of the requestor (e.g., “a member of your community”); things medical researchers do to encourage participation (e.g., provide results); concerns about participating in medical research; attitudes toward medical researchers; health status, health-related quality of life, health behaviors and conditions, and health care use; knowledge of research procedures; and social and demographic characteristics.

Questions about socio-demographic characteristics were asked at the end of the survey using practices consistent with the European Commission’s “Guidance note on the collection and use of equality data based on racial or ethnic origin” (see Table 1) [22]. The respondents were categorized into the four ethnoracial groups based on self-reports to two questions about their ethnoracial identity. The first question asked, “Are you Hispanic or Latino?” The respondents answering “yes” (n = 100) were classified as “Latino” regardless of how they answered a follow-up question about their race. To assess race, the respondents were asked, “Which one or more of the following would you say is your race: White, Black or African American, American Indian, Alaska Native, Asian, or Native Hawaiian or Other Pacific Islander?” The questionnaire also included an “other” category for interviewers to capture categories not listed in the question. The interviewers were instructed to record all the categories offered by the respondent; the respondents provided up to three categories. The respondents were classified as: “White,” if they answered “no” to the question on Hispanic origin and reported no other racial categories (n = 102); “Black,” if they answered “no” to the question on Hispanic origin and reported “Black” only (n = 101) or “Black” and “White” as their race (n = 5); or “American Indian” if they answered “no” to the question on Hispanic origin and reported “American Indian” alone (n = 99) or in combination with one or more other racial categories (n = 3). In addition, one respondent who did not answer the question on Hispanic origin, but reported “American Indian” as their race, was classified as “American Indian.” 

The interviewers verified the respondent’s gender with the question: “Additionally, just to verify, you are (male/female)?” No respondents contradicted the interviewer’s assessment of their gender identity. The respondents were collapsed into three educational attainment groups “high school or less”, “some college”, and “college graduate or more” based on self-reports to the question “What is the highest grade or year of school you completed?” Age was assessed by the question, “In what year were you born?”.

### 2.2. Measures

#### 2.2.1. Medical Research Literacy

We measured MRL using questions from a five-item knowledge scale adapted from Brody et al. [1] (see Table 2). The questions were preceded by the introduction: “For the next questions, I am going to read a statement. Please tell me if you feel the statement is: definitely true, mostly true, neither true nor false, mostly false, or definitely false.” The questions asked about issues concerning informed consent and ethical practices, such as whether a person’s participation must be voluntary and whether medical researchers must keep the information confidential. Each question was presented as a statement about medical research participation, and respondents were asked to indicate if they felt the statements were true or false using the response categories in the introduction. The statements were presented in a random order. 

Responses of “definitely true” and “mostly true” are coded as correct for the questions asking about whether participation in medical research must be voluntary (“Voluntary”) and whether medical researchers must keep information confidential (“Confidential”). By contrast, the questions about the goals of medical care (“Goals”), the main purpose of medical care (“Benefit”), and whether a person must continue participating until a study is done (“Continue”) are reversed, such that the “definitely false” and “mostly false” categories are coded as correct. Consistent with Brody et al. [1], we denote a question as a “true statement” if the correct direction for answering is “definitely true” or “mostly true” versus a “false statement” if the correct direction for answering is “definitely false” or “mostly false” (Table 2). The eleven instances of a respondent not answering an MRL question are coded as incorrect for this analysis. We create a summary score for each respondent based on the number of questions answered correctly. A higher score indicates a higher level of MRL.

#### 2.2.2. Expressed Likelihood to Participate in a Clinical Trial

Our dependent variable is the respondent’s self-assessed likelihood of participating in a clinical trial based on responses to the question: “A clinical trial is a study that tests new drugs or treatments. If a medical researcher asked you to participate in a clinical trial, how likely would you be to participate: very likely, somewhat likely, neither likely nor unlikely, somewhat unlikely, or very unlikely?” (see Table 1).

For the analysis, the responses were collapsed into whether the respondent selected either of the first two categories (“very” or “somewhat likely”) versus not. This decision was motivated by existing research which shows that the predictive strength of both “unlikely” responses is similar [23]. Two respondents who did not answer this question are excluded from analyses using this variable.

#### 2.2.3. Experienced Racial Discrimination by a Medical Care Provider

Some previous research indicates a negative relationship between a person’s likelihood to participate in a clinical trial and their self-reports of having experienced discrimination by a medical care provider [24]. The respondents in the survey were asked a yes–no question about past experiences of discrimination by health care providers: “Do you feel you have ever been treated unfairly by a health care provider because of your race or ethnicity?” (see Table 1). One respondent did not answer this question. Reporting a “yes” response varied dramatically across the ethnoracial groups, with levels ranging from 42.5% for the Black respondents, 25.3% for the Latino respondents, 42.2% for the American Indian respondents, and 2.0% for the White respondents. We examine the relationship between this variable and MRL and control for past experiences of racial discrimination by medical care providers in multivariable analyses.

### 2.3. Analytic Strategy

The data analysis was performed in Stata 17. The categorical variables include ethnoracial identification, gender, education, binary expressed likelihood to participate in a clinical trial, past experience of racial discrimination by a health care provider, and responses to the individual MRL questions. Descriptive information on these variables is presented in both absolute (number of respondents) and relative frequencies (percentages). The continuous variables include age and the number of MRL questions correctly answered, which are presented with a mean and standard deviation. (Note that while age is modeled as a continuous measure in regression analyses, we present age in quartiles for descriptive analyses.) The analyses that use continuous outcomes are based on linear regression analyses using regress; analyses using binary outcomes are based on logistic regression analyses using logistic. Pairwise comparisons use pwcompare, which computes results by changing the baseline category, as opposed to controlling for multiple comparisons. 

## 3. Results

### 3.1. Response to Questions about MRL

Table 3 summarizes responses to the MRL questions, both individually (Panel A) and overall (Panel B). The results are shown separately for the “True” and “False” statements. In Panel A, the correct answers for each question are bolded and underlined. Results show that each of the two true statements—Voluntary and Confidential—are correctly identified by over 91% of the respondents, while each of the three false statements—Goals, Benefit, and Continue—are correctly identified by less than 30% of the respondents. This pattern led to over 84% of the respondents getting both true questions right, while almost half of the respondents got none of the false questions right—with only 7.3% getting all three right. This systematic difference in the responses to the true–false statements motivated us to explore whether there were other differences in how respondents answered the true–false statements and how these statements were associated with other outcomes.

### 3.2. Socio-Demographic Characteristics and Correctly Answering MRL Questions

Table 4 and Table 5 present the results examining the relationship between providing a correct answer to each of the MRL questions and respondents’ socio-demographic characteristics and self-reports of having experienced medically-related discrimination. The results are shown separately for the “True” and “False” statements. For descriptive purposes, the percentages of respondents providing a correct answer to the given question for each predictor are provided in Table 4; Table 5 tests for the significance of these patterns by showing odds ratios and 95% confidence intervals from bivariate logistic regression models of providing a correct answer to the given question on each of the socio-demographic characteristics and the indicator for past discrimination.

The results from Table 4 and Table 5 indicate that Black respondents answer the question about voluntary participation (Voluntary) correctly less often than the other racial/ethnic groups. However, Voluntary is the only question for which there are differences among the ethnoracial groups. The results also highlight a strong, positive relationship between education and correct answers, but only among the False Statements. Older age is associated with lowered odds of answering Continue correctly. There are no differences in correctly answering the questions based on gender or self-reported past experience of medically-related discrimination. Overall, Table 5 suggests that different factors may influence false statements.

The relationship between the total number of correct answers (evaluated as a score across the five MRL questions) and the predictor variables is provided in Table 6. The results are shown separately for “All Statements,” “True Statements” (only), and “False Statements” (only). For each set of results, Table 6 presents both the mean and standard deviation of the number of correct answers by the predictor variables, as well as the results from a multivariable linear regression model containing all the predictor variables. The number of correct answers was treated as a continuous outcome.

Turning to the results for “All Statements,” we find that more educated and younger respondents appear to have, on average, better comprehension of essential elements of informed consent and research participation, as reflected in the higher numbers of correct answers they provide to the MRL questions, net of the other socio-demographic characteristics and experience of medically-related discrimination. We find no effect of ethnoracial identification or gender on overall MRL scores when controlling for the other variables in the model.

Based on the results from Table 5, Table 6 further examines the associations separately for the “True” and “False” statements. The results in Table 6 show that the effects of education and age on correct answers seem to only exist for the set of questions written as false statements. As highlighted by results in Table 5, the significant effect of ethnoracial identification on the true statements is largely driven by the lower number of correct answers by Black respondents to the voluntary participation question. Paralleling the bivariate results for the individual questions in Table 5, there are no significant effects based on gender or reports of medically-related discrimination.

### 3.3. Expressed Likelihood to Participate in a Clinical Trial

The first column of results in Table 7 shows the percentage of respondents for each of the predictor variables who reported being “very” or “somewhat likely” to participate in a clinical trial. Under “Bivariate Models,” we show results from bivariate logistic regression models that regress being “very” or “somewhat likely” (versus “neither likely nor unlikely,” “somewhat unlikely,” or “very unlikely”) to participate in a clinical trial on each of the individual predictor variables. These results provide valuable information about the association between expressed likelihood to participate in a clinical trial and the predictor variable before controlling for other variables. Three sets of models are presented under the results for “Multivariable Models.” Model 1 shows results from a model that includes all of the socio-demographic variables and the indicator for experiencing past medically-related discrimination. Models 2a and 2b build on Model 1 by adding the number of correct answers to the MRL statements to the models. Models 2a and 2b differ from each other in that Model 2a uses the number of correct answers to all MRL questions, while 2b enters two different predictors: the number of correct answers to the true statements and the number of correct answers to the false statements.

The bivariate models present an interesting pattern of results. Compared to Black respondents, the odds of expressed likelihood to participate in a clinical trial are lower for each of the other ethnoracial groups, but the relationships are only significant for the Latino and American Indian respondents compared to Black respondents. The odds of expressed likelihood to participate are also significantly lower for respondents with more education. The effect of ethnoracial identification is reduced in the multivariable models, with only the comparison between American Indian and Black respondents remaining significant in Models 2a and 2b. By contrast, education remains significantly associated with lower odds of the expressed likelihood of participating in a clinical trial across the multivariable models. Gender and having experienced past medically-related discrimination are not related to clinical trial participation in any of the models.

Turning to the results for MRL, the bivariate model for the full set of questions (“All Statements”) shows a strong, significant relationship between MRL and expressed likelihood to participate in a clinical trial: respondents are significantly less likely to say they are “very” or “somewhat” likely to participate in a clinical trial with increasing levels of MRL. This relationship holds in Model 2a after controlling for the other predictor variables. Further inspection of this relationship, as shown in bivariate models and Model 2b, suggests that this effect is largely driven by responses to the false statements. Across the models, correct answers to the true statements are associated with higher levels of expressed likelihood to participate; however, the relationships fail to reach statistical significance. By contrast, correct answers to the false statements are significantly and negatively related to expressed likelihood to participate.

## 4. Discussion

Overall, MRL scores are problematically low. Over half of the respondents correctly answered just two or fewer questions out of five, while only 20% correctly answered four or more. Our set of questions covers important, fundamental topics in the understanding of the research and informed consent process. These scores indicate a potentially widespread fundamental misunderstanding of these processes.

We now review how MRL relates to socio-demographics, comparing our findings to existing research on health and science literacy, as this study represents a novel foray into MRL. While ethnoracial identification did not affect overall MRL, when contrasting with Allum and colleagues’ [9] finding on science literacy, we did observe Black respondents incorrectly answering the question on voluntary participation at significantly higher rates. This fundamental misunderstanding likely stems at least in part from historical violations of trust between medical researchers and the Black community, most notably the Tuskegee syphilis study [25,26]. We find that gender does not affect MRL, in accordance with Kiliç et al. [27] and He et al. [28]. Education is a significant positive predictor of MRL, just as Kiliç and colleagues found, while age is a significant negative predictor of MRL, just as He and colleagues found. We observe that both the education and age effects are largely driven by their predictive power among the false rather than true questions.

Next, we find in this diverse sample that, as MRL increases, the expressed likelihood to participate in a clinical trial decreases. This finding is troubling for the reputation of and participation in medical research and suggests the need for serious further inquiry into why those who know more about medical research are less likely to participate. Suggested directions for future exploration into this relationship are discussed later in this section. Given the time and effort that researchers have invested into understanding what goes into the decision to participate or not in a clinical trial, this constitutes an important finding.

Conventional wisdom suggests that the presentation of some questions on a scale as false is a rigorous mechanism to reduce acquiescence bias (agreeing regardless of content) and inattention in surveys. However, we find that respondents are more likely to answer MRL questions correctly when the correct answer is “true” rather than “false.” As described in the methods section, we conducted cognitive interviews prior to the survey administration to uncover any problems participants demonstrated comprehending the content of the MRL statements. These interviews did not indicate that the content of the false statements was more complex than the true statements. Thus, the evidence presented here suggests that the pathway to the significantly reduced correct response rates for the false statements seems to be solely the result of the grammatical and conversational differences of the reversed response categories. This added methodological artifact should be considered carefully in future research that seeks to operationalize MRL and related knowledge-based constructs.

Research into trust in medical researchers and its relationship with medical research literacy is a second direction we believe deserves deeper investigation. While it was not the object of analysis in this study, it is very likely an intervening variable in the relationship between MRL and the likelihood of participating in medical research. Tsai et al. [29] found a positive relationship between health literacy and trust in physicians and the healthcare system, so it is plausible that such a relationship exists in the medical research sphere as well. We believe that a more rigorous measurement of MRL is first required before such deeper analyses can be reliably executed, but the theoretical considerations and significant bivariate effects found in this study suggest the potential for important findings for this line of research.

One limitation of this study is that the outcome of expressed likelihood to participate in a clinical trial is just that—expressed. This contrasts with an actual decision to participate in a clinical trial, which is of more interest to researchers. It is plausible, if not likely, that there is a disconnect between expressed and actual likelihood to participate in a clinical trial. Acquiescence bias may play a role in inflating the reported likelihood to participate, as well as the respondent’s knowledge that the proposed clinical trial is hypothetical. Furthermore, an actual invitation and decision to participate would likely be affected by the specific circumstances of the invitation, the content of the letter, the identity of the researcher(s), and many other factors that are not considered when answering our hypothetical question about participation. The effect of the identity of the researchers was something we were able to investigate. As discussed in Section 2.2, the Voices Heard survey also contained questions asking respondents how likely they would be to participate in a medical research study, depending on who made the request. Preliminary analyses of the responses to these questions indicate dramatic variations of trust in various individuals and institutions. Echeverri et al. [30] corroborated these findings, writing that “participants were more willing to participate… in studies led by their own healthcare providers, and local hospitals and universities.” The identity of the individual or institution requesting participation clearly matters, and an ambiguous medical researcher, as appears in our question, may not elicit the same trust as a local institution. However, we are forced to consider the expressed likelihood of participating in a hypothetical clinical trial as a sufficient proxy for the actual decision. 

Finally, while the Voices Heard sample is notably balanced on many socio-demographics, it should be considered that it is still somewhat small (n = 410) and comprised entirely of Wisconsinites. Thus, the findings must be generalized with caution. Further research is recommended to replicate our results both with larger, more representative samples and with more geographically wide-reaching populations.

## 5. Conclusions

The implications of medical research literacy are exciting and warrant further research. As alluded to above, the limitations of this study’s evidence prevent a universal relationship between MRL and participation in medical research from being taken for granted, but with replicated results, we are optimistic that medical research literacy can be an important insight into why individuals do or do not participate in medical research studies. Furthermore, unexpected results regarding knowledge-based statements written so that a response of “false” represents a correct answer should be considered in future research, and an explanation or disproval of the phenomenon is desirable. By continuing this line of investigation, we hope to increase the representation of historically underrepresented groups in medical research and ensure that the benefits of its valuable contributions to public health are enjoyed by all.

## Figures and Tables

**Table 1 ijerph-19-15168-t001:** Descriptive Statistics of Respondents’ Socio-demographic Characteristics, Expressed Likelihood to Participate in a Clinical Trial, and Experienced Past Racial Discrimination.

Variables	n	Percent or Mean and (Standard Deviation)
Socio-demographic Characteristics		
Ethnoracial Identification		
Black	106	25.9%
Latino	100	24.4%
American Indian	102	24.9%
White	102	24.9%
Gender		
Male	142	34.6%
Female	268	65.4%
Education		
High School or Less	127	31.0%
Some College	135	32.9%
College Grad or More	148	36.1%
Age	410	44.9 (16.9)
Expressed Likelihood to Participate in a Clinical Trial		
Very Likely	66	16.2%
Somewhat Likely	121	29.7%
Neither Likely nor Unlikely	38	9.3%
Somewhat Unlikely	79	19.4%
Very Unlikely	104	25.5%
Missing	2	
Experienced Past Discrimination		
Yes	115	28.1%
No	294	71.9%
Missing	1	

**Table 2 ijerph-19-15168-t002:** Exact Wording of the Medical Research Literacy Questions.

Question Label	Question Statement	Correct Direction
Voluntary	A person’s participation in medical research must be voluntary.	True
Confidential	Medical researchers must keep information about participants in their studies confidential.	True
Goals	The goals of regular medical care are the same as the goals of medical research.	False
Benefit	A main purpose of medical research is to provide a direct benefit to the person in the study.	False
Continue	If you agree to participate in a medical research study, you must continue participating until the study is done.	False

**Table 3 ijerph-19-15168-t003:** Frequency Distribution of Answers to MRL Questions.

Panel A: Frequency Distribution by Question										
	True Statements				False Statements					
	Voluntary		Confidential		Goals		Benefit		Continue	
Response Categories	n	%	n	%	n	%	n	%	n	%
Definitely true	** 264 **	** 64.7 **	** 253 **	** 62.2 **	55	13.6	51	12.6	129	31.6
Mostly true	** 111 **	** 27.2 **	** 117 **	** 28.8 **	135	33.3	125	30.8	113	27.7
Neither true nor false	17	4.2	19	4.7	102	25.1	112	27.6	56	13.7
Mostly false	10	2.5	11	2.7	** 72 **	** 17.7 **	** 77 **	** 19 **	** 47 **	** 11.5 **
Definitely false	6	1.5	7	1.7	** 42 **	** 10.3 **	** 41 **	** 10.1 **	** 63 **	** 15.4 **
Correct	375	91.9	370	91	114	28.1	118	29.1	110	27
Incorrect	33	8.1	37	9	292	71.9	288	70.9	298	73
Missing	2		3		4		4		2	
**Panel B: Frequency Distribution of Total Number of Correct Answers**										
		**True Statements**			**False Statements**			**All Statements**		
**Correct Answers**		**n**	**%**		**n**	**%**		**n**	**%**	
0		11	2.7		204	49.8		1	0	
1		53	12.9		100	24.4		24	5.9	
2		346	84.4		76	18.5		198	48.3	
3		---	---		30	7.3		104	25.4	
4		---	---		---	---		60	14.6	
5		---	---		---	---		23	5.6	

**Table 4 ijerph-19-15168-t004:** Percentage of Respondents Providing a Correct Answer to MRL Questions by Socio-demographic Characteristics and Experienced Past Discrimination.

	True Statements		False Statements		
Predictors	Voluntary	Confidential	Goals	Benefit	Continue
Socio-demographics					
Ethnoracial Identification					
Black	83.0	86.8	26.7	29.5	25.5
Latino	97.0	90.9	26.3	25.2	31.3
American Indian	93.1	94.1	28.4	30.3	26.7
White	95.1	92.1	31.0	31.0	24.5
Gender					
Male	88.7	90.8	22.1	27.7	24.8
Female	93.6	91.0	31.4	29.8	28.1
Education					
High School or Less	89.7	89.8	14.3	16.7	19.7
Some College	94.0	92.5	26.7	32.3	26.3
College Grad or More	91.9	90.5	41.4	36.7	33.9
Age (in quartiles)					
1st Quartile: 18–31	91.8	90.0	25.5	25.5	37.3
2nd Quartile 31–44	88.5	88.4	34.4	32.6	28.1
3rd Quartile: 44–57	90.6	94.3	25.5	34.0	20.6
4th Quartile: 57–90	96.9	90.6	27.7	24.2	21.1
Experienced Past Discrimination					
No	92.2	90.1	26.6	29.6	27.0
Yes	91.2	92.9	32.2	28.1	27.2

**Table 5 ijerph-19-15168-t005:** Bivariate Logistic Regression Models of Providing a Correct Answer to the Individual MRL Questions on the Predictor Variables.

	True Statements				False Statements					
	Voluntary		Confidential		Goals		Benefit		Continue	
Predictors	Odds Ratio	95% CI	Odds Ratio	95% CI	Odds Ratio	95% CI	Odds Ratio	95% CI	Odds Ratio	95% CI
Socio-demographics										
Ethnoracial Identification										
Black	-	-	-	-	-	-	-	-	-	-
Latino	**6.55** **	**1.86–22.98**	1.52	0.63–3.69	0.98	0.53–1.83	0.81	0.43–1.50	1.33	0.73–2.45
American Indian	**2.75** *	**1.09–6.89**	2.41	0.89–6.54	1.09	0.59–2.01	1.04	0.57–1.89	1.07	0.57–1.99
White	**3.97** **	**1.41–11.14**	1.77	0.71–4.42	1.24	0.67–2.26	1.07	0.59–1.94	0.95	0.51–1.78
Gender										
Male	-	-	-	-	-	-	-	-	-	-
Female	1.88	0.92–3.85	1.02	0.50–2.08	1.59	0.99–2.57	1.11	0.71–1.75	1.18	0.74–1.89
Education										
High School or Less	-	-	-	-	-	-	-	-	-	-
Some College	1.81	0.72–4.53	1.40	0.59–3.32	**2.18** *	**1.16–4.09**	**2.39** **	**1.32–4.32**	1.46	0.81–2.61
College Grad or More	1.31	0.57–2.97	1.08	0.49–2.40	**4.24** ***	**2.33–7.71**	**2.90** ***	**1.63–5.17**	**2.08** **	**1.20–3.62**
Age	1.02	0.99–1.04	1.01	0.99–1.03	1.00	0.98–1.01	0.99	0.98–1.01	**0.98** **	**0.96–0.99**
Experienced Past Discrimination										
Yes	-	-	-	-	-	-	-	-	-	-
No	0.89	0.41–1.93	1.44	0.64–3.26	1.31	0.82–2.10	0.93	0.58–1.50	1.01	0.62–1.65
**Pairwise Comparisons**										
Ethnoracial Identification										
American Indian vs. Latino	0.42	0.11–1.67	1.58	0.54–4.62	1.12	0.60–2.08	1.29	0.70–2.40	0.80	0.43–1.48
White vs. Latino	0.61	0.14–2.61	1.16	0.43–3.15	1.26	0.68–2.34	1.33	0.71–2.47	0.71	0.38–1.32
White vs. American Indian	1.44	0.44–4.71	0.73	0.25–2.20	1.13	0.62–2.07	1.03	0.57–1.87	0.89	0.47–1.67
Education										
College Grad or More vs. Some College	0.72	0.28–1.82	0.77	0.33–1.81	**1.94** *	**1.17–3.21**	1.22	0.74–1.99	1.43	0.85–2.39

* *p* < 0.05, ** *p* < 0.01, *** *p* < 0.001.

**Table 6 ijerph-19-15168-t006:** Multivariable OLS Regression Models of MRL Scores (Correct Answers) on the Predictor Variables by All Statements, True Statements Only, and False Statements Only.

	All Statements				True Statements				False Statements			
Predictors	Mean	S.D.	Coef (S.E.)	*p*	Mean	S.D.	Coef (S.E.)	*p*	Mean	S.D.	Coef (S.E.)	*p*
Socio-demographics												
Ethnoracial Identification												
Black	2.52	0.98	-		1.70	0.57	-		0.81	1.02	-	
Latino	2.68	0.95	0.04 (0.14)	ns	1.86	0.40	**0.19 (0.07)**	**	0.82	0.94	−0.17 (0.14)	ns
American Indian	2.71	1.01	−0.04 (0.14)	ns	1.85	0.38	**0.17 (0.07)**	*	0.85	0.96	−0.21 (0.14)	ns
White	2.72	1.02	0.20 (0.15)	ns	1.86	0.40	**0.17 (0.07)**	*	0.85	1.00	0.04 (0.15)	ns
Gender												
Male	2.52	0.91	-		1.78	0.51	-		0.74	0.92	-	
Female	2.72	1.02	0.15 (0.10)	ns	1.84	0.42	0.06 (0.05)	ns	0.88	1.00	0.08 (0.10)	ns
Education												
High School or Less	2.30	0.75	-		1.79	0.47	-		0.50	0.82	-	
Some College	2.69	0.98	**0.35 (0.12)**	**	1.84	0.42	0.02 (0.06)	ns	0.84	0.94	**0.34 (0.12)**	**
College Grad or More	2.93	1.08	**0.61 (0.12)**	***	1.82	0.47	−0.02 (0.06)	ns	1.11	1.04	**0.64 (0.12)**	***
Age	-	-	**−0.01 (0.00)**	*			0.00 (0.00)	ns			**−0.01 (0.00)**	**
Experienced Past Discrimination												
Yes	2.64	1.00	-		1.82	0.46	-		0.82	0.98	-	
No	2.69	0.97	0.05 (0.11)	ns	1.82	0.43	0.03 (0.05)	ns	0.87	0.98	0.03 (0.11)	ns
**Pairwise Comparisons**												
Ethnoracial Identification												
American Indian vs. Latino	-	-	−0.05 (0.14)	ns	-	-	−0.01 (0.06)	ns	-	-	−0.04 (0.14)	ns
White vs. Latino	-	-	0.20 (0.16)	ns	-	-	−0.01 (0.08)	ns	-	-	0.21 (0.16)	ns
White vs. American Indian	-	-	0.24 (0.16)	ns	-	-	0.00 (0.08)	ns	-	-	0.24 (0.16)	ns
Education												
College Grad or More vs. Some College	-	-	**0.26 (0.12)**	*	-	-	−0.05 (0.05)	ns	-	-	**0.30 (0.11)**	**

ns = not significant, * *p* < 0.05, ** *p* < 0.01, *** *p* < 0.001.

**Table 7 ijerph-19-15168-t007:** Logistic Regression Models of Expressed Likelihood to Participate in a Clinical Trial on Socio-Demographics and Medical Research Literacy.

				Multivariable Models					
		Bivariate Models		Model 1		Model 2a		Model 2b	
Predictors	%	Odds Ratio	95% CI	Odds Ratio	95% CI	Odds Ratio	95% CI	Odds Ratio	95% CI
Socio-demographics									
Ethnoracial Identification									
Black	58.5	-	-	-	-	-	-	-	-
Latino	41.4	**0.50** *	**0.29–0.87**	0.61	0.33–1.11	0.60	0.33–1.10	0.56	0.30–1.03
American Indian	37.6	**0.43** **	**0.25–0.75**	0.55	0.30–1.00	**0.54** *	**0.30–0.99**	**0.50** *	**0.27–0.93**
White	45.1	0.58	**0.34–1.01**	0.82	0.43–1.57	0.85	0.44–1.64	0.80	0.42–1.56
Gender									
Male	51.4	-	-	-	-	-	-	-	-
Female	42.9	0.71	0.47–1.07	0.79	0.50–1.23	0.81	0.52–1.27	0.80	0.51–1.25
Education									
High School or Less	62.7	-	-	-	-	-	-	-	-
Some College	44.4	**0.48** **	**0.29–0.78**	**0.52** *	**0.31–0.86**	**0.56** *	**0.33–0.94**	**0.56** *	**0.33–0.95**
College Grad or More	32.7	**0.29** ***	**0.18–0.48**	**0.31** ***	**0.18–0.52**	**0.35** ***	**0.20–0.60**	**0.36** ***	**0.21–0.62**
Age	-	1.00	0.99–1.02	1.00	0.99–1.02	1.00	0.99–1.01	1.00	0.98–1.01
Experienced Past Discrimination									
Yes	44.0	-	-	-	-	-	-	-	-
No	50.0	1.27	0.82–1.96	1.46	0.89–2.40	1.49	0.90–2.46	1.48	0.89–2.45
MRL Correct Answers									
All Statements	-	**0.70** ***	**0.57–0.86**			**0.79** ***	**0.63–0.98**		
True Statements Only	-	1.12	0.73–1.73					1.12	0.70–1.81
False Statements Only	-	**0.67** ***	**0.54–0.83**					**0.75** *	**0.59–0.94**
**Model Statistics**									
Likelihood Ratio Chi-Square				34.54		39.05		41.98	
Log Likelihood				−263.34		−261.08		−259.62	
Degrees of freedom				8		9		10	
n				407		407		407	

* *p* < 0.05, ** *p* < 0.01, *** *p* < 0.001.

## Data Availability

Data supporting the results will be stored as anonymized data and metadata form in DRYAD. DRYAD is an organization that archives data for public access, and the authors’ institution is a DRYAD member. This service is free and is compliant with the new NIH data availability rules.

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
