# Peer review of "Exploring the Relationship between Medical Research Literacy and Respondents’ Expressed Likelihood to Participate in a Clinical Trial"

_ijerph, 2022, doi:10.3390/ijerph192215168_

Round 1

Reviewer 1 Report

The manuscript report critical and methodologically consistent results of collecting and using socio-demographic caracteristics and frecuency distribution of expressed likelihood to participate in a clinical trial. The approach seems solid, statistically sound and temporary. It fits very well with the five main publick health challenges to watch in the Post-Pandemic Era (Syed Hamza Sohail, 2022). The coincidence is evident at least on the second of five chalenges, recognized world wide and referring to the modernization of digital transformation practices of achieving data for different communities. Notwithstanding, there are two big problems envisioned in this manuscript. The Tables are not easy to understand, and the part of these looka illy arranged. And sometimes, the standard deviation is mentioned but not presented (Line 148, Table 1). It is strongly encouraged to re-adopt all Tables to the general Journal format.  Table 6 is increadibly time-consuming to understand. But, the main concern is collecting and using data based on the racial or ethnic origin. The extreme suggection is to reference and re-analyze carefully the "Guidance Note on the Collection and Use of Equality data based on racial or ethnic origin", published in 2021 (Luxemburg, Pub. Office of the EU) by the Subgroup on Equality Data of High-Level Group on Non-discrimination, Equality, and Diversity (Luxemburg, Pub. Office of the EU). The last piece of advice refers primarily to the Discusion (4) and  Coclusion (5) Sections. Some evident statements are better to exclude, calling attention to the most substantial and interesting facts based on this research. 

Reviewer 2 Report

I want to congratulate the authors on this important study. The paper has merit and will be of great interest to the research/health community.

Firstly, I wanted to comment on two concerns I had with the MRL survey questions. Using the binary Male and Female appears to be very dated. Why not use an alternative use as ‘other’?

Secondary you state that the correct answer to the question on goals is false. But it is possible they could be aligned. I would have thought a neutral response was more appropriate.

My first concern was the assertion that MRL and health literacy are co-related. The authors observed irregular results and now consider that the confounding factor of trust may be at play. This is a sound assumption. I did not see any evidence of the testing of the MRL tool within these 4 groups to ensure the wording was understood as intended. High level terms are used in the survey that may have difference meanings within the various groups. I would like to see the authors address this issue.

I would suggest that further development of a survey tool be co-design using a grounded approach to address the cultural differences between the groups.

Given that some of the 4 groups have different experiences of health care and therefore ‘likeliness’ will be influenced by past trauma, I think this needs to be dealt with before you can draw outcomes from this research.

Round 2

Reviewer 1 Report

The paper is ready for publication. Notwithstanding, it still has one stong incongruence in Section 4, "Discussion" (lines 382-383). The observation that:" As the content of the false atatements is not particularrly complex than the true statements, the pathway to their significantly reduced correct response rates....." has no experimental or corresponding references sustenance. The seminal papers of H. McClusky (1934, " The Negative Suggestion Effect of the False Statements in the True-False Test") and Thomas C. Toppino and H. Ann Brochin (1989, " Learning  from Tests: The case of True-False Examinatiob"), among the others, can be taken to support the STRONGEST and PROBABLY MORE CRUCIAL empirical result of the research under review. These data can further improve the more general Lee et al. (2017) methodological approach ("Introduction to data-driven Methodologies for Prognostics and Health Management"). 

Author Response

Reply is uploaded as a word/pdf file
